# Activity-induced polar patterns of filaments gliding on a sphere

Chiao-Peng Hsu [1,2], Alfredo Sciortino [1,2], Yu Alice de la Trobe[1] & Andreas R. Bausch [1✉]

Active matter systems feature the ability to form collective patterns as observed in a plethora of living systems, from schools of fish to swimming bacteria. While many of these systems move in a wide, three-dimensional environment, several biological systems are confined by a curved topology. The role played by a non-Euclidean geometry on the self-organization of active systems is not yet fully understood, and few experimental systems are available to study it. Here, we introduce an experimental setup in which actin filaments glide on the inner surface of a spherical lipid vesicle, thus embedding them in a curved geometry. We show that filaments self-assemble into polar, elongated structures and that, when these match the size of the spherical geometry, both confinement and topological constraints become relevant for the emergent patterns, leading to the formation of polar vortices and jammed states. These results experimentally demonstrate that activity-induced complex patterns can be shaped by spherical confinement and topology.

[1] Center for Protein Assemblies and Lehrstuhl für Zellbiophysik (E27), Physics Department, Technische Universität München, Garching, Germany. [2]These authors contributed equally: Chiao-Peng Hsu, Alfredo Sciortino. ✉email: abausch@mytum.de

L iving matter exhibits a captivating range of collective beha-
viors offering insight into how complex patterns can emerge
at different length scales[1]. Examples of living active matter
are found from macroscopic schools of fish[2] to microscopic motile
bacteria and in cell migration[3–5] up to organoid structures[6–9] Such
collective motion is also observed in the realm of colloidal active
materials[10]. Despite the diverse nature of these systems, they all
display interactions that favor the alignment of neighboring
constituents, thus leading to similar forms of collective motion[11].
The structural complexity of the emerging patterns is typically
dictated by the combination of alignment mechanisms between
the constituent building blocks and their shape.

Gliding assays, in which cytoskeletal filaments are driven by
molecular motors grafted to a substrate, are an ideal candidate to
address the microscopic mechanisms that underlie pattern for-
mation in active systems. Specifically, due to their high aspect
ratio and intrinsic directionality, gliding filaments represent an
example of polar active particles. Collective patterns of these
cytoskeletal systems on a solid substrate with weak to moderate
interactions include polar waves[12–14], nematic lanes[14,15], and
vortices[16]. Recently, actin filaments driven on a liquid membrane
have been shown to display locally polar structures despite a
microscopic regime of nematic binary interactions[17].

While most of these gliding assays have been studied in a two-
dimensional, planar geometry, a spherical topology can also have a
non-trivial impact on the organization of collective systems.
General properties of spherical geometries include that they are
curved, closed, and no defect-free vector field can be defined on the
surface. Specifically, the Poincaré-Hopf theorem dictates that the
total topological charge on a sphere has to be $+2$[18,19]. For these
reasons, surface curvature and topology are known to play a role in
the observed collective patterns in several systems, for instance,
during the growth of the corneal epithelium[20] or embryonic
development[21]. Yet, only a limited number of experimental[9,22],
theoretical[23,24], and simulation[25–30] studies has been carried out
under spherical constraint. In the case of polar particles moving
persistently in one direction, one common result of these studies is
the formation of a state characterized by a polar band that rotates
along the equator, which is the geodesic in spherical geometry.

In this work, we present the formation of polar patterns in a
system of actin filaments gliding on the inner leaflet of a vesicle.
As the concentration varies, we observe not only the often pre-
dicted emergence of a polar band but also several diverse such as
off-equator polar vortices and jammed patterns. Our setup
represents an experimental realization of a system of polar fila-
ments on spherical geometry. The possibility to observe the
dynamic process leading to pattern formation thus allows us not
only to connect the microscopic interaction between filaments to
their collective behavior but also to show what is the specific
impact of the spherical geometry on the resulting structures.

## Results

**Collective patterns on active actin vesicles**. We perform gliding
assays in spherical geometry by encapsulating actin filaments (F-
actin), biotinylated heavy meromyosins (bHMM) motors, and
streptavidin inside vesicles (mean radius of $\approx 18\,\mu$m) using the
continuous droplet interface crossing encapsulation (cDICE)
method[31,32] ("Methods", Fig. 1a). The length of fluorescent
F-actin is modulated by gelsolin to a log-normal distribution with
mode length $L \approx 0.6\,\mu$m (i.e., much shorter than the vesicles'
radius) in our experiments ("Methods", Supplementary Fig. 1).
The bHMM motors bind via streptavidin to the inner leaflet of
the vesicles containing 5 mol% biotinylated lipids. Non-adsorbing
polymer poly(ethylene glycol) (PEG; 0.6 wt%) is used to push
F-actin on the inner leaflet of the vesicle by depletion forces.

The membrane-bound bHMM motors, fueled by energy from
adenosine triphosphate (ATP) hydrolysis, propel the F-actin on
the inner leaflet of the vesicle. An ATP-regeneration system and
an oxygen-scavenging system are also included.

The main consequence of the fluidity of the lipid bilayer is that
motors slip on it while pushing the filaments, which hinders their
ability to exert forces and reduces the activity of the propelled
filaments[17,33]. This results in strong steric interactions being
enforced between filaments as, differently than on glass substrates,
motors are unable to push filaments on top of each other. Instead,
when a filament collides with another filament, it must stop, and
only eventually, by bending its tip, can resolve the collision by
aligning (Fig. 1b), as already described[17,33]. In this context, the
anisotropy of the filaments and the presence of a lower friction
coefficient along their main axis also play a role in that only one of
the two filaments changes direction[34]. As opposed to nematic
structures being observed at a high-activity regime where filaments
have the tendency to align but can still slide on top of each
other[14,35], this low-activity stop-and-go interaction, on a planar
geometry, has recently been shown to result in the formation of
polar patterns, both in experiments and in theoretical works[17,35].

We image 100–300 vesicles for each encapsulated actin
concentration ($c_A$) 60 min after the production by epifluorescence
microscopy ("Methods"). We observe that most of the filaments
separate in space and assemble into patterns on the membrane of
the vesicles in a concentration-dependent manner (Fig. 1c). At $c_A =$
100 nM, the vesicles start displaying streams of actin filaments,
which are elongated structures composed of motile filaments
resulting from multiple collisions on the membrane[17]. At higher
actin concentrations, ring-like structures ($c_A = 300$ nM) and more
packed, complex patterns ($c_A = 600$ nM) can be observed.
Conversely, the vesicles show no patterns if any of the streptavidin-
biotin-interaction components is removed from the system
(Supplementary Fig. 2). Based on Onsager's model for the two-
dimensional system of passive hard rods, the isotropic-nematic
transition occurs at a critical surface density $\rho^* = 3\pi/2L^2$, where $L$ is
the length of the rods[36]. While $\rho^* \approx 15.6$ filaments/$\mu$m$^2$ is expected
for our actin filament system, a surface density $\rho \approx 1.8$–10.7
filaments/$\mu$m$^2$ (corresponding to surface packing fraction $\phi =$
0.007–0.04) for $c_A = 100$–600 nM (Supplementary Information)
already leads to the formation of patterns. The fact that collective
patterns emerge at $\rho < \rho^*$ confirms that the patterns are activity
induced[37,38].

**Polar patterns emerge on the spherical surface**. To visualize the
patterns on the vesicles' surface, we use the equirectangular
projections of the vesicles, which transform spherical coordinates
into planar ones[39]. By examining individual vesicles at different
$c_A$, we identify five polar patterns: streams (Fig. 2a), vortices
(Fig. 2b, c), partially jammed patterns (Fig. 2d), bands (Fig. 2e),
and globally jammed patterns (Fig. 2f). The probability of
observing a given pattern depends on $c_A$. Streams are the dom-
inating polar pattern at $c_A = 100$ nM, with the probability of their
observation decreasing as $c_A$ is increased (Fig. 2g). Vortices are
ring-like, defect-free polar patterns composed of aligned filaments
that loop at a latitude between the equator and the pole of the
vesicles and are most dominant in the middle concentration
regime of $c_A = 300$ nM. Bands are defined as polar patterns that
span over the surface, circulate at the equator, and are often
observed at higher concentrations. At higher surface coverage, the
low-force regime of the molecular motors also leads to the for-
mation of partially and globally jammed patterns in which fila-
ments are unable to move as they impede each other's motion.

The fact that different polar patterns emerge with increasing $c_A$
(Fig. 2g) confirms that streams act as the building blocks for more

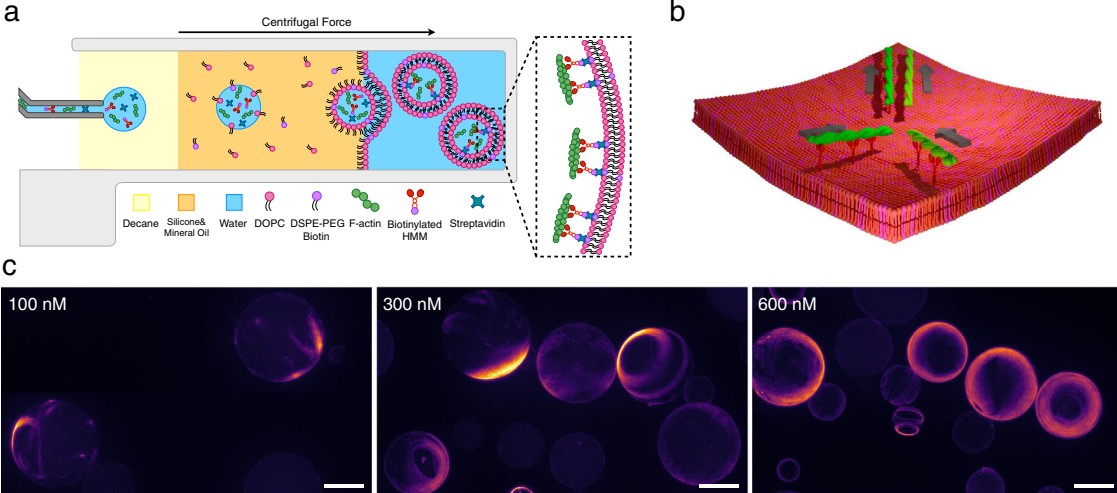

**Fig. 1 Encapsulation of active actin filaments in giant unilamellar vesicles. a** Schematic of the vesicle production process. The protein mixture containing F-actin, biotinylated HMMs, and streptavidin is encapsulated inside giant unilamellar vesicles (GUVs). **b** Schematic of a collision of two actin filaments on the inner leaflet of a GUV. The two filaments have a strong tendency to align after the collision due to the stop-and-go interaction. **c** Fluorescence images (z-projections) of GUVs with encapsulated actin concentration at 100, 300, and 600 nM. Scale bars, 20 μm.

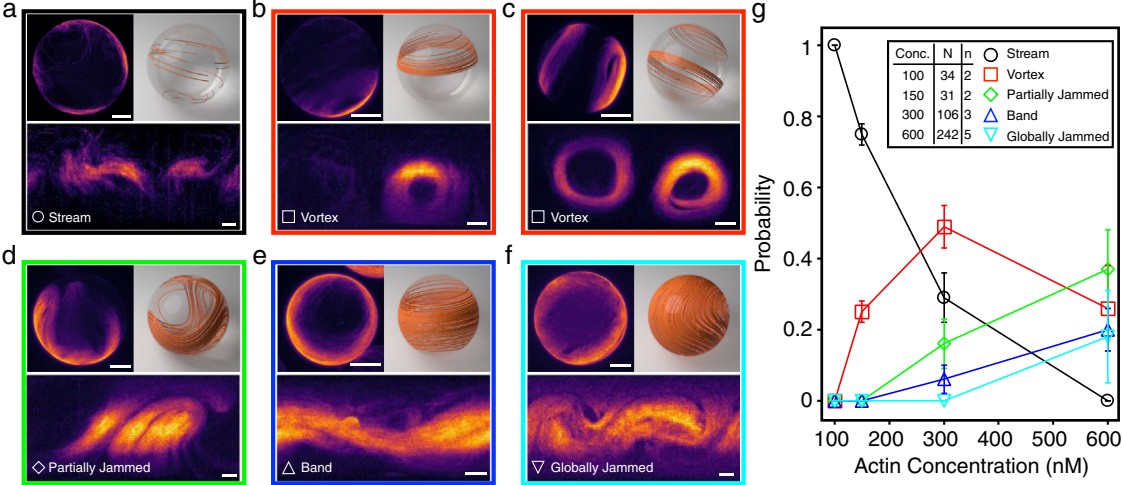

**Fig. 2 Polar patterns on active actin vesicles. a–f** Z-projections (top-left), 3D schematics (top-right), and equirectangular projections (bottom) of vesicle displaying different polar patterns: stream (**a**), vortex (**b**), vortices (**c**), partially jammed (**d**), band (**e**), and globally jammed (**f**). Scale bars, 10 μm. **g** Distributions of each observed polar pattern at different encapsulated actin concentrations. Inset indicates the number $n$ of repeated experiments and the total number $N$ of vesicles counted at each concentration. Error bars represent the standard deviations from repeated experiments. Empty vesicles are excluded from the statistics. The graph shows a tendency towards more complex structures as the concentration increases and topological constraints become more relevant.

complex structures on the closed surface. Depending on the concentration, the structures either cover only a fraction of the vesicle's surface or the full vesicle, but show no dependence on the vesicles' radius (Supplementary Fig. 4).

Patterns that cover the full vesicle, such as bands and jammed states, must satisfy the Poincaré-Hopf theorem[18,19] and display a total topological charge of +2. They differ in that bands display two +1 defects, and jammed states instead show multiple +1/2 defects. On the other hand, streams, vortices, and partially jammed states are not bound by this constraint. We classify them differently only because they, while still arising from streams as fundamental blocks, display qualitatively different behaviors that have to do with confinement rather than with topology.

**Role of stream interaction on pattern formation.** To gain insight into the emergence of actin patterns within the vesicles,

we observe their dynamics over time by confocal microscopy. We see that individual filaments powered by motors glide and collide with each other on the membrane in a steric repulsion-dominated manner[17]. Judging from confocal movies, the system appears to be in a steady state after ≈15–30 min from the beginning of the experiment and is active for at least another 30 min in the presence of an ATP-regeneration system. The tendency of filaments to move together for a prolonged time when colliding polarly and the fact that filaments stop upon collision introduce an asymmetry at the binary collision level, stabilizing polar pairs with respect to anti-polar ones[40].

Multiple filament collisions on the membrane thus lead to the buildup of streams already at low surface densities ($\rho \approx 1.8$ filaments/μm², $\phi = 0.007$) (Supplementary Movie 1). The individual streams explore the inner surface of the vesicle at a velocity of 40–50 nm/s (Fig. 3a and Supplementary Movie 2).

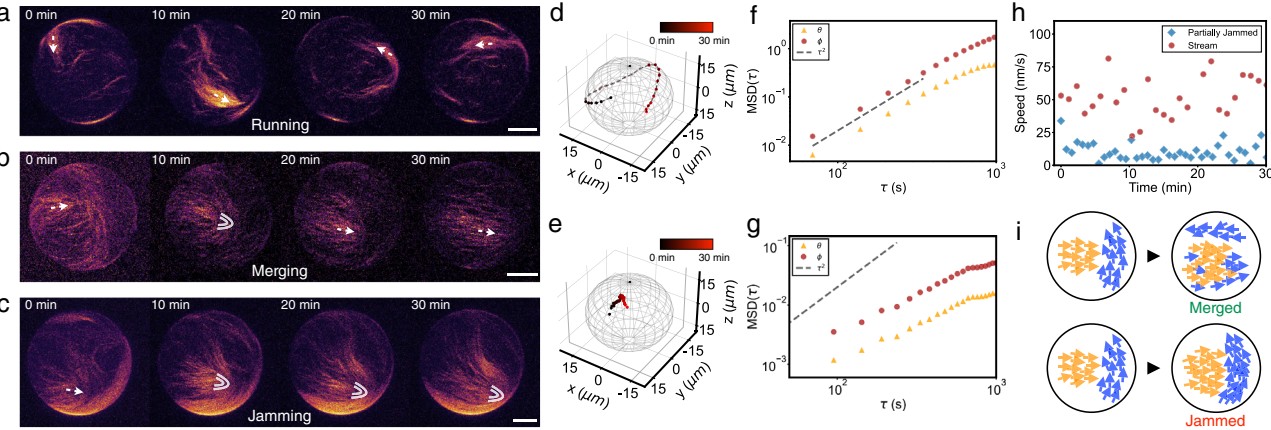

**Fig. 3 Stream interaction on an active actin vesicle. a–c** Time series (z-projection of confocal stacks) of different individual vesicle displays a stream running at $c_A = 100$ nM (**a**), two streams merging at $c_A = 150$ nM (**b**), and two streams jamming at $c_A = 300$ nM (**c**). Arrows (white) indicate the direction of motion. Parabolic symbols (white) indicate the +1/2 topological defects. Scale bars, 10 μm. **d** Trajectories of the stream from (**a**) moving on a vesicle. **e** Trajectories of the +1/2 defect from the partially jammed vesicle (**c**). Time is color-coded; the total time is 30 min. **f** Mean squared displacement (MSD) of the stream with respect to the polar angles $\theta$ (triangle; yellow) and $\phi$ (circle; red). Dashed line indicates ballistic behavior (MSD $\propto \tau^2$). **g** MSD of the +1/2 defect in a partially jammed vesicle. Dashed line indicates ballistic behavior. **h** The instantaneous speed of stream (circle; red) and +1/2 defect (diamond; blue), shown in (**d**) and (**e**). **i** Schematics of streams merged (top) and streams jammed (bottom) on the vesicles.

Streams do not appear to move differently at different positions on the GUVs, although they can slow down and reorient if they hit an obstacle. Thus the speed of filaments is rather dependent on the mean local alignment than on the local concentration or position. Streams interact with each other also in a steric stop-and-go manner, the same as individual filaments do. Different streams align upon collision and have a strong tendency toward polar alignment. By this up-scaling of the interaction, multiple streams interact with each other to build up larger structures at higher filament concentrations.

The collision of two streams with different orientations can also lead to the formation of a transient local +1/2 topological defect (Fig. 3b and Supplementary Movie 3). This transient +1/2 defect, which dissolves when streams merge, plays the role of sorting the streams' polarity which is similar to the polarity sorting mechanism observed on planer lipid membranes[17]. Due to spherical confinement, this merging mechanism results here in the coalescence of multiple streams into a structure spanning the surface of the vesicles. Compared to a flat surface, the collision probability of different streams and thus their merging is increased due to the closed surface of the sphere.

At higher filament density, however, streams do not necessarily merge upon collision but can also hinder each others' motion. This jamming mechanism of two streams can be observed when a local +1/2 topological defect does not dissolve after the collision (Fig. 3c and Supplementary Movie 4). Here, the motion of both streams is slowed down significantly with the jammed streams moving at a velocity of 5–10 nm/s, which is 5–10 times slower than observed for the moving streams. Unlike the merging of streams, such local jamming has been rarely observed on supported lipid bilayers (SLBs) system[17], where collisions were mostly resolved in the time frame of the experiments. This difference arises from the fact that the lipid mobility is expected to be twofold higher in vesicles than in SLBs[41], which leads to more slippage of the motors and thus lower counter-forces for the motors, by which no sufficient forces are exerted on the filaments to overcome the locally jammed structure. This is supported by the observation that jamming occurs mostly when thicker and longer streams interact with each other. Consequently, filaments are confined in the locally jammed, defect-like configuration. By the tracking ("Methods") of both a free moving stream (Fig. 3d)

and a +1/2 defect from a partially jammed vesicle (Fig. 3e), we can visualize their trajectory and compute the mean squared displacement ("Methods", Fig. 3f, g) as well as the instantaneous speed (Fig. 3h). This clearly shows the difference between the two configurations in terms of their motion.

We then identify merging and jamming as the mechanisms for the development of polar patterns on a sphere (Fig. 3i). While the former enables coarsening of structures, the latter conversely slows down the active motion.

**Vortex and band pattern are formed by streams.** As the concentration is increased, merging streams evolve into different structures. At $c_A = 150$ nM, 29% of the vesicles display a vortex and the probability to observe vortices is further increased at $c_A = 300$ nM. Both a single vortex (Fig. 2b) and double vortices (Fig. 2c) can be observed on a vesicle. To elucidate the formation and polarity of vortices, we investigate the time evolution of vortex-forming vesicles (Fig. 4a and Supplementary Movie 5). In the beginning, multiple streams form and explore the vesicle's surface until they collide and merge with another stream. Such streams can build up a polar vortex pattern by forming a closed loop with themselves on the vesicle. Vortices have a speed comparable to that of streams at an average speed of ≈45 nm/s ("Methods", Fig. 4b–d). We also observe that vortices have a variable radius depending on the length of the streams.

The buildup of double vortices requires that the two vortices are either spatially separated on the vesicle (Fig. 2c) or that they have opposite handedness, e.g., two concentric vortices with opposite rotation directions (Fig. 4e and Supplementary Movie 6). In the latter case, two long streams develop simultaneously close to each other, yet with opposite polarity, which prevents them from merging. As the interaction between streams and the probability that an elongated stream loops on itself is enhanced by the finite area of the spherical geometry, the formation of vortices is a direct effect of confinement. Thus multiple streams merging and closed-loop forming due to spherical confinement represent a unique and straightforward approach to build up vortex patterns on a sphere (Fig. 4f). On SLBs, elongated structures have a smaller chance of looping on themselves, and

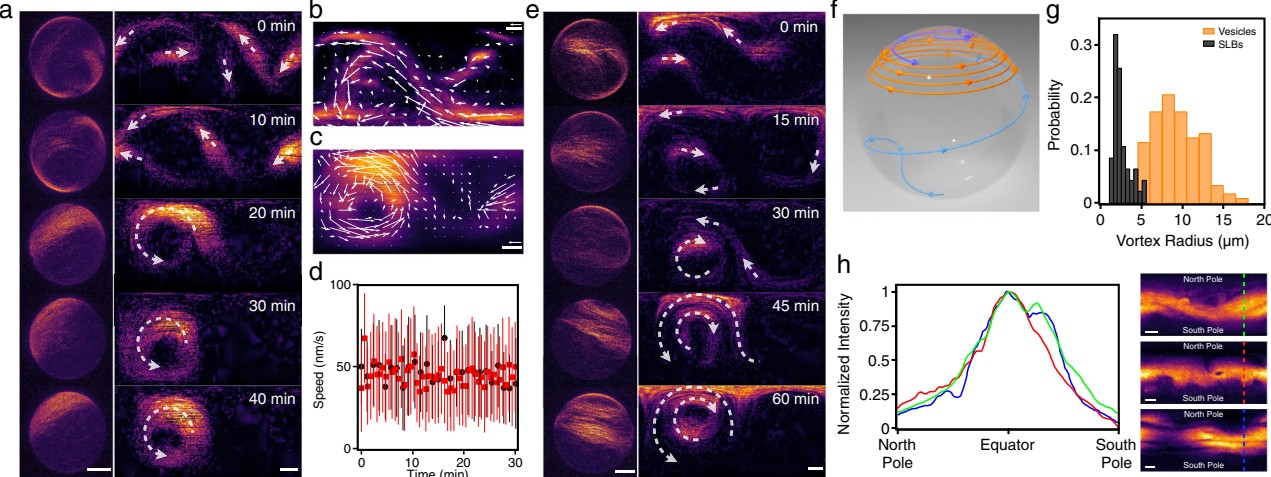

**Fig. 4 Vortex formation on active actin vesicles. a,** Time series of the Z-projections (left) and equirectangular projections (right) show the formation of a vortex. Arrows (white) indicate the direction of streams/vortices. Scale bars, 10 μm. **b–c** Mean intensity projection overtime of the equirectangular projections of a stream at $c_A = 100$ nM (**b**) and a vortex at $c_A = 300$ nM (**c**). Arrows (white) show the averaged optical flow over time. Scale bar, 10 μm. Arrow, 50 nm/s. **d** Averaged instantaneous speed after the formation of stream (circle; black) and vortex (square; red). Error bars represent the standard deviations of the velocity extracted from optical flow. The speed of vortices and streams are found to be consistent with each other at an average speed of ≈45 nm/s (stream: 46 ± 7 nm/s; vortex: 45 ± 6 nm/s). **e** Time series of the Z-projections (left) and equirectangular projections (right) show the formation of double vortices. Arrows (white) indicate the direction of streams/vortices. Scale bars, 10 μm. **f** Schematic of the trajectories of streams during the vortex formation on a vesicle. In this case, the stream from the top hemisphere (blue) merges with the stream from the bottom hemisphere (light-blue) building up the circulating vortex (orange) at the top hemisphere. **g** Distributions of the radius of the vortices on vesicles (orange, bin width = 1.5 μm) and on supported lipid bilayers[17] (black, bin width = 0.5 μm). **h** The normalized intensity profiles of polar bands on a sphere (left). The intensity profiles are obtained from the equirectangular projections (right). Scale bars, 10 μm.

the observed vortices are smaller[17]. The mean radius of vortices is 2.4 μm and 9.0 μm when forming on SLBs and vesicles, respectively (Fig. 4g). Because the size of confinement, of the order of the GUVs' diameter, matches the typical length of polar structures, vortices and bands in this system can be globally polar. While on the SLB system polar order is not long-ranged as different structures have different polarity[17], here confinement allows to select a single definite polarity. This is akin to the formation of polar bands observed in simulations of self-propelled rods with volume exclusion in 2D with boundary conditions and small system size[35].

When the filament concentration is high enough that the length of the stream matches the size of the vesicle's equator, a vesicle-spanning polar band occurs (Fig. 2e and Supplementary Movie 7). Because they cover the whole surface, polar bands are bound by topological constraints of the sphere, which they accommodate by forming two +1 defects at the poles, as required by the Poincaré-Hopf theorem[18,19]. In contrast, vortices explore only a fraction of the vesicle's surface and thus do not require a topological charge of +2 and are indeed defect-free. In both cases, filaments, being aligned, can freely glide inside the structure. Hence bands move with an approximate speed around 40 nm/s and appear to be stable, i.e., not prone to bending instabilities. Bands also feature an intensity distribution which is peaked at the equator and decreases at the poles, where the defects are. This closely resembles the one predicted by previous studies[24] and further hints that band formation originates from a closed, curved topology (Fig. 4h).

**Structure of jammed patterns**. We observe that at high concentrations ($c_A = 600$ nM), the probability of observing vortices decreases, and instead, globally jammed patterns emerge, indicating that the impact of the jamming mechanism increases with increasing actin concentration (Fig. 2g). A globally jammed

vesicle displays four comet-like +1/2 defects to accommodate the required net topological charge of +2 on a sphere. (Fig. 5a, b and Supplementary Movie 8). These +1/2 defects restrict the filaments' motion as each defect structure acts as an obstacle for the movement of the others. Consequently, while vortices and bands exhibit a clear coherent polar rotation, the jammed patterns show no significant motion of the filaments. We track the positions of defects in globally jammed vesicles from confocal images (Fig. 5c). After an initial equilibration, the defects quickly slow down and stop moving almost completely, just slightly fluctuating in position. Defects are found not to move significantly (instantaneous speed ≲ 5 nm/s, Fig. 5d) over at least 30 min, and their mean squared displacement indicates diffusive behavior (Supplementary Fig. 5).

At any given time, the positions of the four +1/2 defects can also be described by $\theta_{ij}$, which denotes the angle between radii from the vesicle center to each of the six defect pairs, $ij$. For a tetrahedral configuration, the mean pair angle $\langle \theta_{ij} \rangle = 109.5°$ and for a planer configuration $\langle \theta_{ij} \rangle = 120°$. It has been demonstrated in an active nematic vesicle that the four +1/2 defects repeatedly oscillate between the tetrahedral and planar configurations and the frequency is set by the activity of the component and the size of the sphere[22,30]. In contrast, the four defects in the globally jammed vesicle (Fig. 5a) evolve very slowly and as a consequence, the mean pair angle $\langle \theta_{ij} \rangle$ does not oscillate and stays stable over time (Fig. 5e). This stable defect configuration is due to the reduction of activity of actin filaments when arrested in a globally jammed state on a sphere. Whether a band or a jammed state form is thus entirely dependent on which way the system chooses to accommodate the fundamental requirement on the total charge. As this four-defect configuration is more likely to form, globally jammed patterns and not bands are the dominant ones at high density and underline the influence a spherical topology can have in shaping collective patterns in this system.

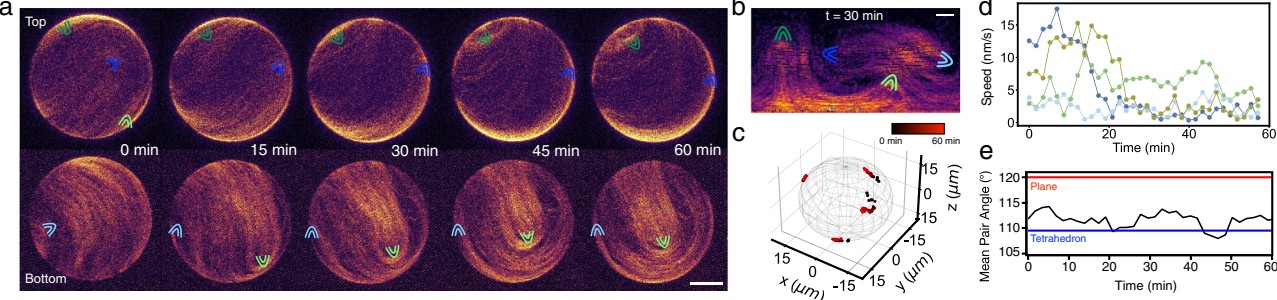

**Fig. 5 Jammed state on active actin vesicles. a** Time series of the hemisphere projections of a globally jammed vesicle where the four +1/2 defects are indicated with parabolic symbols. Both the top and bottom hemispheres are shown separately. **b** The equirectangular projection of the vesicle in (**a**) at $t =$ 30 min where the four +1/2 defects are indicated with parabolic symbols. Scale bars indicate 10 μm. **c** Trajectories of the four +1/2 defects on a globally jammed vesicle. Time is color-coded; the total time is 60 min. The positions of the defects are extracted from confocal images and projected back on the GUV's surface **d** Instantaneous speed of the four +1/2 defects from (**c**). **e** The time evolution of average angular distances of all six defect pairs $\langle \theta_{ij} \rangle$ where red line indicates the planar configuration $\langle \theta_{ij} \rangle = 120°$ and blue line indicates the tetrahedral configuration $\langle \theta_{ij} \rangle = 109.5°$.

## Discussion

To summarize, we have observed activity-induced polar patterns on a sphere in a system of actin filaments gliding on the internal surface of giant lipid vesicles. Our experimental results show that the emergence of vortex patterns at low concentrations and of bands and jammed patterns at high concentrations can be rationalized by the stop-and-go filament interaction, combined with the effect of geometrical confinement and topological constraints. While in this experimental system, the microscopic interaction leading to polar patterns is similar to what is observed on planar SLBs[17], here the combined effect of confinement and topology defines the observed final states of the patterns. The spherical confinement leads even at low filament concentrations $\phi$ to the buildup of unreported vesicle-circulating structures, i.e., polar vortices. The fact that vortices circulate at a latitude different than the equator indicates that the tendency of filaments to move along the sphere's geodesics (great circles such as the equator)[9,24,25] can be suppressed by lateral filament-filament steric interactions, possibly also enhanced by the presence of a depletant (PEG) pushing filaments together, that stabilizes the off-equator vortices. This suggests that to capture this behavior in theoretical models additional terms describing the elongated structure of filaments and their lateral interaction must be taken into account. While the formation of vortices is an effect of confinement, as the density increases, topological constraints play a more significant role by imposing a total nematic charge of +2 and forcing the system to form either a band or a jammed pattern.

Globally jammed states have also been rarely reported[27]. They feature the same configuration of the active nematic defects on vesicles[22], but in this system, filaments are polar and unable to cross each other, which results in frozen defects. Band structures around the equator instead are predicted by simulations of polar spherical particles gliding on a sphere at high surface packing fractions[23,25] and are also described by a theory based on long-ranged hydrodynamic interactions[24,42]. While in nematic systems globally jammed states and bands have been found to transition into each other[30], we never observed such a transition in our system, possibly due to the polar nature of the system and to the low activity of gliding filaments. Despite our system being also based on local interactions in a dry limit where no long-range hydrodynamic interactions are present[43,44] and filaments only interact sterically, current models also do not capture the formation of vortices and jammed states. These can be attributed to the unique stop-and-go interaction and to the high aspect ratio of the filaments, which is lacking in present models.

Moreover, although HMM is a non-processive motor that spends only a fraction of its duty cycle connected to actin, motors' diffusion might also affect the spatial distribution of motors which are expected to accumulate in the presence of high-density patterns. This has already been shown to have a destabilizing effect on the formation of patterns in a similar 2D system[45]. Further experiments on characterizing the spatial distribution of motors in the 3D geometry are needed to confirm if such coupling between the density of motors and filaments is also present in our system.

Understanding the impact of the microscopic details of active agents' interaction, whether local or long-range, in the emergence of collective patterns on curved, confined geometries remains a challenge for experiments and theory alike[40,46]. Yet, as many biological systems live in a closed, curved geometry, rotational motions are emerging as common characteristics in nature. Simple model systems such as the one we present here may shed light on more complex dynamics observed in cellular structures such as those reported for organoids and spheroids[8,9] and on active systems of elongated particles with steric interactions in general. Finally, the scenario presented here may also play an important role in the design and control of new non-equilibrium patterns.

## Methods

**Materials and reagents**. Imidazole, magnesium chloride (MgCl2), potassium chloride (KCl), calcium chloride (CaCl2), sodium azide (NaN3), boric acid (H3BO3), monopotassium phosphate (KH2PO4), dimethyl sulfoxide (DMSO), tris-(hydroxymethyl)-aminomethan (Tris), dithiothreitol (DTT), adenosine triphosphate (ATP), sucrose, glucose, pyranose-oxidase (PO), catalase (C), creatine phosphokinase (CPK), creatine phosphate (CPH), Poly(ethylene glycol) 35 kDa (PEG), methyl cellulose 40 kDa (MC), decane, silicon oil 50 cSt, mineral oil, and chloroform were purchased from Sigma-Aldrich. Alexa Fluor 488 phalloidin (488 phalloidin), phalloidin, and NHS-Biotin were purchased from Thermo Fisher Scientific. 1,2-dioleoyl-sn-glycero-3-phosphocholine (DOPC) and 1,2-distearoyl-sn-glycero-3-phosphoethanolamine-N-[biotinyl(polyethylene glycol)-2000] (DSPE-PEG(2000) Biotin) were purchased from Avanti Polar Lipids.

**Buffers**. G-actin buffer (G-buffer) (pH 8.0) contains 2 mM Tris, 0.2 mM ATP, 0.2 mM CaCl2, 0.2 mM DTT, and 0.8 mM NaN3. Assay buffer (A-buffer) (pH 7.4) contains 25 mM Imidazole, 4 mM MgCl2, 25 mM KCl, and 1 mM EGTA. Assay-Tris buffer (AT buffer) (pH 7.4) contains 50 mM Tris, 4 mM MgCl2, 25 mM KCl, and 1 mM EGTA. F25-buffer (pH 7.5) contains 50 mM Tris, 2 mM MgCl2, 0.5 mM ATP, 0.2 mM CaCl2, and 25 mM KCl. Myosin buffer (M-buffer) contains 0.6 M KCl, 10 mM KH2PO4, and 2 mM DTT.

**Proteins**. Actin and skeletal muscle myosin were purified from rabbit skeletal muscle[47,48]. No rabbits were directly involved in the study. Monomeric actin (G-actin) was stored in G-buffer at 4 °C. Heavy meromyosin (HMM) was prepared by

dialyzing ground rabbit skeletal muscle against the M-buffer at 4 °C. Biotinylated HMM (bHMM) was prepared by incubating HMM with NHS-biotin in a 1:10 molar ratio in DMSO at room temperature for 10 min (Supplementary Information). Gelsolin was purified from adult bovine serum (Sigma-Aldrich) (Supplementary Information). Streptavidin was purchased from Thermo Fisher Scientific. Bovine serum albumin (BSA) was purchased from Sigma-Aldrich.

**Assay mixture preparation.** Non-labeled actin filaments (F-actin) were obtained by incubating 20 μM G-actin and 10 μM phalloidin in F25-buffer at room temperature for 30 min. Fluorescent-labeled F-actin were obtained by incubating 5 μM G-actin, 50 nM gelsolin, and 2.5 μM 488 phalloidin in F25-buffer at room temperature for 30 min to achieve the mode length $L \approx 0.6\,\mu m$ (Supplementary Information). Enzymatically inactive bHMMs were eliminated by centrifuging ($350,000 \times g$, 25 min) 2 μM bHMM, 10 μM non-labeled F-actin, 1 mM DTT, and 2 mM ATP in A-buffer at 4 °C. Afterward, bHMM was incubated on ice with streptavidin in a 1:1 molar ratio in A-buffer for 20 min. The assay mixture contains 100–600 nM fluorescent-labeled F-actin, 150 nM bHMM, 150 nM streptavidin, 1 mM DTT, 4 mM ATP, 18 U/mL CPK, 9 mM CPH, 8 U/mL PO, 1.7 kU/mL C, 36 mM glucose, 100 mM sucrose, and 0.6 wt% PEG in AT buffer. In this mixture, PO and C function as an oxygen-scavenging system to prevent protein denature and photobleaching during fluorescence imaging. CPK and CPH serve as an ATP-regeneration system. The assay mixture was prepared as the inner phase before the start of vesicle production.

**Vesicle production.** Vesicles were produced using the continuous droplet interface crossing encapsulation (cDICE) method[31]. Briefly, it consisted of a cylindrical rotating chamber, successively filled with a glucose solution to collect the vesicles, a lipid-in-oil solution to saturate the oil/water interfaces, and decane as the continuous phase in which droplets were produced. The rotating chamber (inner diameter: 70 mm; top opening diameter: 30 mm; collecting opening diameter: 8 mm; height: 2 mm.) was 3D-printed (Clear Rasin, Form 2; Formlabs). The osmolality of the glucose solution was prepared to be 10–20 mOsm/kg higher than the encapsulated inner phase. The lipid-in-oil solution contained 0.5 mM of DOPC/DSPE-PEG(2000) Biotin (19:1, M/M) in a decane/silicon oil/mineral oil (3:40:7, v/v) mixture as previously described[49]. The chamber was successively filled with 1.5 mL glucose solution, 3 mL lipid-in-oil solution, and 1.5 mL decane while rotating at 1800 rpm. The assay mixture was injected from a glass capillary by inserting the capillary's tip (40 μm inner diameter) in the decane. The injection was regulated by a pressure pump (AF1; Elveflow) so that the vesicles were produced in a span of 3 min. The vesicles were collected from the cDICE chamber after the production.

**Imaging and data acquisition.** An equal volume of the vesicle solution and 1% MC solution (osmolality-matched) were mixed to increase the outer phase viscosity in order to prevent drifting during imaging. Well chambers (sticky-Slide 8 Well; ibidi) and glass chambers that consist of cover-slips (Carl Roth) fixed to microscope slides (Carl Roth) by 4-layer parafilm were used for imaging. Both imaging chambers were passivated with a BSA solution (10 mg/mL) for 20 min before the insertion of the vesicles. The vesicles were imaged immediately after the production to record the formation of polar patterns using glass chambers sealed with vacuum grease (Bayer Silicones) and 60 min after the production to observe the developed polar patterns using well chambers. Leica Thunder Imaging System with an HCX PL APO 63×/1.40 oil immersion objective and Leica TCS SP5 confocal microscope coupling resonant scanner with an HCX PL APO 63×/1.40 CS2 oil immersion objective were used to image the vesicles. The equirectangular projections of a single vesicle were produced using the Map3-2D software[39]. Defect identification was performed manually from confocal z-projection images using Fiji/ImageJ.

**Manual tracking of structures.** We tracked defects and streams from equirectangular projections and confocal projections by marking the position on the plane of the core of a defect or of the center of mass of a stream and the reprojecting it back on a sphere according to the following maps, for which the radius $R$ of the considered GUV must be known. For an equirectangular projection, the position $(\phi, \theta)$ maps to a position $\mathbf{r} = R\big(\cos(\phi)\sin(\theta),\ \sin(\phi)\sin(\theta),\ \cos(\theta)\big)$, whereas for a confocal projection, given the point $(x, y)$ on the plane, then $\mathbf{r} = \left(x,\ y,\ \sqrt{R^2 - x^2 - y^2}\right)$.

The speed was obtained as the arc-length on the sphere between two consecutive positions, divided by the time interval, i.e., $v = \frac{R}{\Delta t}\cos^{-1}(\mathbf{r}_{i+1}\cdot\mathbf{r}_i/R^2)$ where $\Delta t$ is the time interval between frames. Manual tracking result is found to be consistent with the one obtained by optical flow method (Supplementary Fig. 6).

**Angular mean squared displacement.** The angular mean squared displacement (MSD) is computed from the angular position $(\phi, \theta)$ at two time-points separated by an interval $\tau$ has been computed as $\langle(\theta(t+\tau) - \theta(t))^2\rangle$ and $\langle(\phi(t+\tau) - \phi(t))^2\rangle$. Their behavior is still is expected to scale (sub)linearly in time for (sub)diffusive motion and quadratically in time for ballistic motion at short times, whereas at long time the behavior saturates[50].

**Optical flow analysis.** To obtain the optical flow, we used a custom-made Python3 script based on OpenCV's sparse optical flow function[51] which tracks the features on images. The script was run on equirectangular projections over time of selected vesicles. This allowed to obtain the angular position $0 \le \theta_i \le \pi$ and $0 \le \phi_i \le 2\pi$ of selected features over each time-frame $i$. To obtain the speed of structures, the following map from polar to Cartesian coordinates was used $\mathbf{r}_i = R\big(\cos(\phi_i)\sin(\theta_i),\ \sin(\phi_i)\sin(\theta_i),\ \cos(\theta_i)\big)$, where $R$ is the GUV's radius. From this, the speed was obtained as the arc-length on the sphere divided by the time interval, i.e., $v = \frac{R}{\Delta t}\cos^{-1}(\mathbf{r}_{i+1}\cdot\mathbf{r}_i/R^2)$, where $\Delta t$ is the time interval between frames.

## Data availability

Microscopy data that support the findings of this study are available in Zenodo at https://doi.org/10.5281/zenodo.5783872. All other relevant data supporting the findings of this study are available within the article and its Supplementary Information files or from the corresponding author upon reasonable request. Source data are provided with this paper.

## Code availability

The code used for analyzing the data of this study is available in Zenodo at https://doi.org/10.5281/zenodo.5783872.

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

## Acknowledgements
We gratefully acknowledge financial support by the the Deutsche Forschungsgemeinschaft (DFG, German Research Foundation), Project number 111166240 SFB863 (B1) and the European Research Council (ERC) under the European Union's Horizon 2020 research and innovation program (grant agreement no. 810104-PoInt).

## Author contributions
A.R.B., A.S., C-P.H., and Y.A.T. designed the research, performed the research, and analyzed the data. A.R.B, A.S., and C-P.H. wrote the manuscript. All authors reviewed and revised the manuscript.

## Funding

## Competing interests
The authors declare no competing interests
