## [Peer review file · Nature Communications]

REVIEWER COMMENTS

Reviewer #1 (Remarks to the Author):

The authors study experimentally gliding assays consisting of actin filaments and meromyosin motors inside a vesicle. A series of collective effects are reported to occur on the inner layer of the spherical vesicle as the actin concentration is varied. The authors observe the formation of polar bands (or streams), the formation vortices, and jammed configurations. The topology of the vesicle plays a central role.

The manuscript is well written and the results clearly presented. More importantly, the reported observations and performed analysis are of timely interest.

Below I list some points that could be improved/discussed.

i. The reported phenomenology seems to be very similar to the one observed in self-propelled rod systems. Early studies of self-propelled rods predicted, already 16 years ago, the emergence of polar clusters despite the fact that binary interactions (in diluted systems) lead to nematic alignment. Coarse-grained, hydrodynamic equations in active matter have failed to account for this observation. However, these theoretical approaches have not been called into question, and it has been argued that this is not a major issue since in simulations it was reported that polar order is not long-ranged (or QLRO). But in a finite system with periodic boundary conditions such polar bands can be system spanning, see Fig. 2e of Grossmann et al. Nat. Comm. 11, 5356 (2020). In my opinion, the vortex observed by the authors on the vesicle results from a similar mechanism and the geometry of the vesicle allows the polar band to get connected (i.e. the head and tail of the band) as occurs in simulations with periodic boundary conditions in the mentioned reference. I think the system analyzed by the authors provides empirical evidence of this stabilizing effect.

In addition to that, Grossmann et al. 2020 discuss a central issue in the authors' current manuscript: the difference between gliding assays on glass and on lipid layers (that is the focus of Sciortino and Bausch PNAS 2021). In the provided reference, it is formally shown how the symmetry of the emergent order is affected by activity -- see Fig 4 of the above reference -- showing that when the rods can glide on top of each other (at high activity) the emergent structures are nematic, while when crossing is not allowed (at low activity) the emergent structures are polar.

Finally, one aspect that is often not discussed is, that beyond the anisotropic shape of rods (which is central to understanding torques in the system), there is also the fact that rods can exhibit a distinct

friction coefficient: the resistance that exerts the rod as we try to move it along its major axis is significant less than the one is experienced as we try to move it in the perpendicular direction, or when we try to rotate it. Thus, by playing with the mobility tensor of the rod, it is possible to prevent displacements in the direction perpendicular to the major axis so that in a T-shape collision between rods (as in Fig. 1B of Sciortino and Bausch PNAS 2021), only one of the rods rotates until getting aligned with the other rod. This is discussed in the above given reference (as well as in Weitz et al. PRE 92, 012322, 2015).

ii. I presume that the formation of vortices at different latitudes can be strongly affected by the speed of the filaments (and thus to their activity) as well as actin concentration. The speed of the filaments and its potential correlation with the position (i.e. latitude) and/or concentration has not been reported. Furthermore, the authors discussed jamming, but there is no information on the speed of filaments or their mean-square displacement. In particular to characterize jamming, these quantities can be very informative.

iii. The authors do not comment on the spatial distribution of motors on the observed patterns and in particular on jamming states. If I understand correctly, motors can glide on the lipid layer, and thus a coupling between the density of motors and filaments may be present and may play a fundamental role on the emergent dynamics.

Reviewer #2 (Remarks to the Author):

This manuscript presents an experimental study of actin filaments confined to move on the inner leaflet of a giant lipid vesicle. This is a major step forward in having a simple, elegant, and easily tuneable experimental realisation of an active system confined to curved geometry and non-trivial topology. Results are very convincing, and the paper is clearly written and easy to follow.

I have, however, two questions/comments.

1. I am puzzled by the two-vortex state. If one assumes the standard model of self-propulsion along the direction of polarisation, the motion would be along a great circle (i.e., an equator). This is easy to understand since the great circle is a geodesic, i.e., the curved-space analogue to a straight line. In order to move on a small circle (e.g., fig 2c), one needs to apply additional force normal to the trajectory. This

is due to a small circle having a non-zero geodesic curvature. This is just an analogue of the fact that if one wants to curve a straight path in the flat space, one needs to apply a force normal to the direction of motion. Could authors comment on what would cause such force? The reason this has not been predicted by any of the models is that all models to date assumed self-propulsion along the direction of polarisation.

2. The partially and fully hammed states resemble what has been referred to as the “bending of the band” state in Ref. 30. Could authors please comment about similarities/differences between the two?

Minor points:

1. In line 35, I suggest replacing the somewhat confusing term “nematic charge” with “topological charge”, which is a proxy for a more precise definition of the index of a vector field.

2. In lines 40-41, I think it would be helpful to mention that the equator is a geodesic (i.e., the analogue of a straight line).

3. In line 54, I think it is important to note in the main text that the distribution of filament lengths is log-normal.

4. Related to the previous point, I think it is important to emphasize that filaments are much shorter than the radius of the vesicle but that they form long structures that can reach or exceed the vesicle size. This is already stated but I feel it should be emphasized more clearly. In the first reading, I thought that filaments have lengths comparable to the vesicle radius.

5. Presumably, filaments are not bendable. This should also be mentioned.

Overall, this is a very nice work that will greatly contribute to and, hopefully, inspire further work on the active matter on curved surfaces.

Reviewer #3 (Remarks to the Author):

I have read the manuscript 'Activity-induced polar patterns of filaments gliding on a sphere' by Hsu et al with great interest. The field of active matter on curved surfaces was kickstarted by Keber et al [22] using a microtubule-kinesin system. The active nematic patterns seen there kickstarted a lot of theoretical and experimental interest (as referenced here).

Conversely, nobody (until now) has managed to create an active *polar* system on curved surfaces, at least not at densities with significant collective effects.

In addition to being a significant experimental feat on its own, creating this system will also help clarify a lot of thorny questions about polar vs. nematic activity, dry vs. pairwise active and substrate interactions and the role of the shape of the active agent.

This is a significant experimental advance, and based on that alone I recommend publication.

However, I was quite disappointed with some aspects of the manuscript and I don't think it is ready for publication without more work.

First of all, the analysis of the patterns in Figures 2,3 4 and 5 is very heavily qualitative instead of quantitative.

It has been established (see e.g [22,24,25,30,40]) that the patterns found on these surfaces can be uniquely classified using their topological defects with both half and integer defects present. Here, this has only been done so (by hand) for the 'jammed' state.

Is there no way to extract a polarisation or velocity field from the data? I acknowledge that this is a very hard experiment, but it does look like the authors have 3d information at sufficient temporal resolution to try to obtain velocity information at least. In addition to using it to look for topological defects, it could (potentially more easily) be used to find the speed of the band and vortex states, and the speed reduction in the jammed states.

Second, I found some aspects simply unclear and it is hard to put the results here into context without more precise information:

- The mechanism of activity here is actin filaments being propelled over the inner surface of the vesicle which acts like the substrate in a motility assay. Is this correct - the manuscript is less than clear on this point? The authors need to expand on the quite cryptic 'The membrane bound [motors] non-processively propel the F-actin on the inner leaflet of the vesicle'.

Is this then really a 'dry' active interaction, i.e. non-momentum conserving, as stated in the discussion? The inner membrane is presumably fluid, so I would expect a flow induced by the activity there as well. The authors also emphasise the 'slippage of the motors on the lipid bilayer' - what is happening here?

- I would appreciate if the authors include more information about the experimental state: Is this system in steady-state, or does activity decrease as a function of time? There seems to be a probabilistic phase diagram (Fig 2), which would be a first for these systems.

Is this truly probabilistic, or are there uncontrolled variables here, in particular activity level and vesicle radius (which is quite variable, see Fig. S3)?

- Figure 2 and related analysis: The classification here is done purely qualitatively, and additionally 'only vesicles showing clear patterns are considered'. As mentioned above, I am also unsure why there should be that many coexisting states at all. I don't understand how this classification can be done unambiguously, and there are no error bars on the probability distribution plot.

In particular, vortex and band states are topologically the same, and I don't see how a clear dividing line can be drawn. Also, when does a stream become a vortex?

For the (partially) jammed states, the presence of $\pm 1/2$ defect states could be a good indicator as (implicitly) done here. But it would be far better to also show that they are actually jammed, i.e. that the velocity of the material is (much) lower.

- Figure 3 and analysis: As mentioned, a velocity field beyond the schematic picture in 3d would help a lot here. I did not find the supplementary movies associated to this mechanism particularly convincing.

- Figure 4 and analysis have some lovely quantitative data on the band states, and I agree with the authors that the bands share a lot of properties with their theoretical [24] and numerical [25] analogues. There seems to be a bend instability in the band in Fig. 4e - do you observe this systematically? If yes, do you have information about its wave length?

- Figure 5 and analysis: I appreciate the manual $\pm 1/2$ defect tracking here, and it is certainly clear from the 5c that the dynamics is slow, with none of the oscillations seen in [22] present. Were the authors able to qualitatively establish if the defects are moving at all relative to the background? If yes, are they extensile (forward moving) or contractile (backward moving)? In simulations with (round) particles with similar mixed polar and nematic properties [30], extremely slow dynamics compared to the active

motion itself was found (and it was ultimately extensible). I wonder if the present system is part of the same "activity class".

We thank the reviewers for the careful reading of our manuscript and for the insightful comments, which we address in detail below. We report the original comments in italics and add our response in normal font following the phrase "AUTHOR REPLY". Revised text appears highlighted in the resubmitted manuscript. All figure numbers listed here refer to this document.

Reviewer: 1

Comments:

The authors study experimentally gliding assays consisting of actin filaments and meromyosin motors inside a vesicle. A series of collective effects are reported to occur on the inner layer of the spherical vesicle as the actin concentration is varied. The authors observe the formation of polar bands (or streams), the formation vortices, and jammed configurations. The topology of the vesicle plays a central role.

The manuscript is well written and the results clearly presented. More importantly, the reported observations and performed analysis are of timely interest.

AUTHOR REPLY:

We thank the reviewer for the positive evaluation of our work.

Below, I list some points that could be improved/discussed.

i. The reported phenomenology seems to be very similar to the one observed in self-propelled rod systems. Early studies of self-propelled rods predicted, already 16 years ago, the emergence of polar clusters despite the fact that binary interactions (in diluted systems) lead to nematic alignment. Coarse-grained, hydrodynamic equations in active matter have failed to account for this observation. However, these theoretical approaches have not been called into question, and it has been argued that this is not a major issue since in simulations it was reported that polar order is not long-ranged (or QLRO). But in a finite system with periodic boundary conditions such polar bands can be system spanning, see Fig. 2e of Grossmann et al. Nat. Comm. 11, 5356 (2020). In my opinion, the vortex observed by the authors on the vesicle results from a similar mechanism and the geometry of the vesicle allows the polar band to get connected (i.e. the head and tail of the band) as occurs in simulations with

periodic boundary conditions in the mentioned reference. I think the system analyzed by the authors provides empirical evidence of this stabilizing effect.

AUTHOR REPLY:

We thank the reviewer for pointing this out. We have attempted to make this point by hinting that the formation of polar vortices and bands is an effect of confinement, but we now feel this should be addressed more directly.

We have included the following discussion in the revised manuscript:

”Because the size of confinement, of the order of the GUVs’ diameter, matches the typical length of polar structures, vortices and bands in this system can be globally polar. While on the SLB system polar order is not long-ranged as different structures have different polarity [1], here confinement allows to select a single definite polarity. This is akin to the formation of polar bands observed in simulations of self-propelled rods with volume exclusion in 2D with boundary conditions and small system size [2].”

In addition to that, Grossmann et al. 2020 discuss a central issue in the authors’ current manuscript: the difference between gliding assays on glass and on lipid layers (that is the focus of Sciortino and Bausch PNAS 2021). In the provided reference, it is formally shown how the symmetry of the emergent order is affected by activity – see Fig 4 of the above reference – showing that when the rods can glide on top of each other (at high activity) the emergent structures are nematic, while when crossing is not allowed (at low activity) the emergent structures are polar.

AUTHOR REPLY:

This indeed has been one of the main points of our previous work with this system , where filaments were unable to cross each other [1].

We have now expanded the description of microscopic mechanism in our system and the resulting polar structures in the revised manuscript, better putting these observations into context.

”...The membrane-bound bHMM motors, fueled by energy from adenosine triphosphate (ATP) hydrolysis, propel the F-actin on the inner leaflet of the vesicle. An ATP-regeneration system and an oxygen-scavenging system are also included.

The main consequence of the fluidity of the lipid bilayer is that motors slip on it while pushing the filaments, which hinders their ability to exert forces and reduces the activity of the propelled filaments [1, 3]. This results in strong steric interactions being enforced between filaments as, differently than on glass substrates, motors are unable to push filaments on top of each other. Instead, when a filament collides with another filament, it must stop, and only eventually, by bending its tip, can resolve the collision by aligning (Figure 1b), as already described [1, 3]. In this context, the anisotropy of the filaments and the presence of a lower friction coefficient along their main axis also play a role in that only one of the two filaments changes direction [4]. As opposed to nematic structures being observed at a high-activity regime where filaments have the tendency to align but can still slide on top of each other [2, 5], this low-activity stop-and-go interaction, on a planar geometry, has recently been shown to result in the formation of polar patterns, both in experiments and in theoretical works [1, 2].”

Finally, one aspect that is often not discussed is, that beyond the anisotropic shape of rods (which is central to understanding torques in the system), there is also the fact that rods can exhibit a distinct friction coefficient: the resistance that exerts the rod as we try to move it along its major axis is significant less than the one is experienced as we try to move it in the perpendicular direction, or when we try to rotate it. Thus, by playing with the mobility tensor of the rod, it is possible to prevent displacements in the direction perpendicular to the major axis so that in a T-shape collision between rods (as in Fig. 1B of Sciortino and Bausch PNAS 2021), only one of the rods rotates until getting aligned with the other rod. This is discussed in the above given reference (as well as in Weitz et al. PRE 92, 012322, 2015).

AUTHOR REPLY:

This is indeed a very interesting and important point raised by the reviewer, and we have extended the description of the microscopic mechanism in our system in the revised manuscript as suggested by the reviewer above.

ii. I presume that the formation of vortices at different latitudes can be strongly affected by the speed of the filaments (and thus to their activity) as well as actin concentration. The

speed of the filaments and its potential correlation with the position (i.e. latitude) and/or concentration has not been reported.

AUTHOR REPLY:

We have now addressed more head-on the speed of filaments and describe more in detail the movement of structures on the GUVs' surface.

First of all, we generally assume the system behaves similarly to our previous work [1], where the speed of the filaments is found to depend much more strongly on the relative alignment of filaments than on the concentration. We then expect bands and vortices to not have dramatically different speeds and we do not expect spatial variations of the speed along the latitude/longitude. To prove this we recorded the optical flow on equirectangular projections of streams and vortices, from which the speed of structures on the sphere can be extracted as shown in Figure 1a&b. The speed of vortices and streams are found to be consistent with each other (Figure 1c) at an average speed of ≈ 45 nm/s (stream: 46 ± 7 nm/s; vortex: 45 ± 6 nm/s).

We have now specified this in the revised manuscript and added Figure 1 to the Supplementary Information:

"Streams do not appear to move differently at different positions on the GUVs, although they can slow down and reorient if they hit an obstacle. Thus the speed of filaments is rather dependent on the mean local alignment than on the local concentration or position."

"Vortices have a speed comparable to that of streams at an average speed of ≈ 45 nm/s."

Furthermore, the authors discussed jamming, but there is no information on the speed of filaments or their mean-square displacement. In particular to characterize jamming, these quantities can be very informative.

AUTHOR REPLY:

Unfortunately, tracking individual filaments in this system is not feasible given the available resolution. To better characterize the dynamics of jammed vesicles as suggested by the reviewer, we now performed additional analysis. We manually tracked defects and extracted not only their relative angle but also their position and behavior over time. It now appears clear that, after an initial equilibration time, defects stop moving almost completely, as

FIG. 1. **a**, Mean intensity projection overtime of the equirectangular projections of a stream at $c_A = 100$ nM. Arrows show the averaged optical flow over time. **b**, Mean intensity projection overtime of the equirectangular projection of a vortex at $c_A = 300$ nM. Arrows show the averaged optical flow over time. **c**, Averaged instantaneous speed after the formation of stream (circle; black) and vortex (diamond; red). Error bars represent the standard deviations of the velocity extracted from optical flow.

shown in Figure 2a. This is also proven quantitatively by their instantaneous speed and by the mean-squared-displacement (MSD) of their polar angles that show clear non-ballistic behavior (Figure 2b&c). The instantaneous speed of defects is found to be less than 5 nm/s, close to our detection margin. Overall, in the jammed case, defects do not change relative position but also, more strikingly, do not move at all.

The following paragraph and Figure 2 has been added to the revised manuscript:

”We track the positions of defects in globally jammed vesicles from confocal images. After an initial equilibration, the defects quickly slow down and stop moving almost completely, just slightly fluctuating in position. Defects are found not to move significantly (instantaneous speed $\lesssim 5$ nm/s) over at least 30 min, and their mean squared displacement indicates diffusive behavior.”

iii. The authors do not comment on the spatial distribution of motors on the observed patterns and in particular on jamming states. If I understand correctly, motors can glide

FIG. 2. **a**, Trajectories of the four $+1/2$ defects on a globally-jammed vesicle. **b**, Speed of the four $+1/2$ defects. **c**, Mean-square-displacement (MSD) of the four $+1/2$ defects with respect to the polar angles θ (i.e., $\langle(\theta(t+\tau) - \theta(t))^2\rangle$) and ϕ (i.e., $\langle(\phi(t+\tau) - \phi(t))^2\rangle$). Dashed line indicates diffusive behavior ($\text{MSD} \propto \tau$).

on the lipid layer, and thus a coupling between the density of motors and filaments may be present and may play a fundamental role on the emergent dynamics.

AUTHOR REPLY:

Unfortunately, given the 3D geometry of the system and the available resolution, characterizing the spatial distribution of motors is not easily done. The 2D system might shed light on its effect, as it has already recently been shown by Memarian et al. [6]. We have commented on the possible role of motors' diffusion in the revised manuscript:

”Moreover, although HMM is a non-processive motor that spends only a fraction of its duty cycle connected to actin, motors' diffusion might also affect the spatial distribution of motors which are expected to accumulate in the presence of high-density patterns. This has already been shown to have a destabilizing effect on the formation of patterns in a similar 2D system [6]. Further experiments on characterizing the spatial distribution of motors in the 3D geometry are needed to confirm if such coupling between the density of motors and filaments is also present in our system.”

Reviewer: 2

Comments:

This manuscript presents an experimental study of actin filaments confined to move on the inner leaflet of a giant lipid vesicle. This is a major step forward in having a simple, elegant, and easily tuneable experimental realisation of an active system confined to curved geometry and non-trivial topology. Results are very convincing, and the paper is clearly written and easy to follow.

AUTHOR REPLY:

We thank the reviewer for the positive evaluation of our work.

I have, however, two questions/comments.

1. I am puzzled by the two-vortex state. If one assumes the standard model of self-propulsion along the direction of polarisation, the motion would be along a great circle (i.e., an equator). This is easy to understand since the great circle is a geodesic, i.e., the curved-space analogue to a straight line. In order to move on a small circle (e.g., fig 2c), one needs to apply additional force normal to the trajectory. This is due to a small circle having a non-zero geodesic curvature. This is just an analogue of the fact that if one wants to curve a straight path in the flat space, one needs to apply a force normal to the direction of motion. Could authors comment on what would cause such force? The reason this has not been predicted by any of the models is that all models to date assumed self-propulsion along the direction of polarisation.

AUTHOR REPLY:

We thank the reviewer for raising this point which also has us puzzled as it has never been shown. We attribute the off-geodesic motion to two main factors: strong lateral steric interactions, which stabilize structures looping on themselves; the presence of a depletant (PEG) that pushes filament against each other and might further stabilize structures and avoid that filaments "escape" along the geodesic. Current models do not always include such low-activity/high-steric interaction regimes and also usually focus on spherical particles. At the same time, the peculiar elongated structure of filaments, with a high aspect ratio and point forces due to motors distributed along the filament length, might play an additional role.

We have now added this comments in the revised manuscript:

”The fact that vortices circulate at a latitude different than the equator indicates that the tendency of filaments to move along the sphere’s geodesics (great circles such as the equator) [7–9] can be suppressed by lateral filament-filament steric interactions, possibly also enhanced by the presence of a depletant (PEG) pushing filaments together, that stabilizes the off-equator vortices. This suggests that to capture this behavior in theoretical models additional terms describing the elongated structure of filaments and their lateral interaction must be taken into account.”

2. The partially and globally jammed states resemble what has been referred to as the “bending of the band” state in Ref. 30. Could authors please comment about similarities/differences between the two?

AUTHOR REPLY:

Although the partially and globally jammed states display $+1/2$ topological defects as shown in the “bending of the band” state in Ref. 30 in the manuscript. We do think the two things are different for the following reasons: the “bending of the band” arises at high activity (whereas here the system is at low-activity regime), in a nematic system (whereas here it is polar), and is connected to the band to four-defects configuration transition (which we never observed).

To clarify the issue, we have added the following sentence in the revised manuscript:

”While in nematic systems globally jammed states and bands have been found to transition into each other [10], we never observed such a transition in our system, possibly due to the polar nature of the system and to the low activity of gliding filaments.”

Minor points:

1. In line 35, I suggest replacing the somewhat confusing term “nematic charge” with “topological charge”, which is a proxy for a more precise definition of the index of a vector field.

AUTHOR REPLY:

We thank the reviewer’s suggestion and have modified the text accordingly.

2. In lines 40-41, I think it would be helpful to mention that the equator is a geodesic

(i.e., the analogue of a straight line).

AUTHOR REPLY:

We thank the reviewer's suggestion and have modified the text accordingly.

3. In line 54, I think it is important to note in the main text that the distribution of filament lengths is log-normal.

AUTHOR REPLY:

We thank the reviewer's suggestion and have modified the text accordingly.

4. Related to the previous point, I think it is important to emphasize that filaments are much shorter than the radius of the vesicle but that they form long structures that can reach or exceed the vesicle size. This is already stated but I feel it should be emphasized more clearly. In the first reading, I thought that filaments have lengths comparable to the vesicle radius.

AUTHOR REPLY:

We thank the reviewer's suggestion and have modified the text accordingly.

5. Presumably, filaments are not bendable. This should also be mentioned.

AUTHOR REPLY:

Despite their short length, actin filaments are able to bend and this has been shown to facilitate alignment [1].

Overall, this is a very nice work that will greatly contribute to and, hopefully, inspire further work on the active matter on curved surfaces.

AUTHOR REPLY:

We thank the reviewer again for the overall positive evaluation of our work.

Reviewer: 3

Comments:

I have read the manuscript 'Activity-induced polar patterns of filaments gliding on a sphere' by Hsu et al with great interest. The field of active matter on curved surfaces was kickstarted by Keber et al [22] using a microtubule-kinesin system. The active nematic patterns seen there kickstarted a lot of theoretical and experimental interest (as referenced here).

*Conversely, nobody (until now) has managed to create an active *polar* system on curved surfaces, at least not at densities with significant collective effects.*

In addition to being a significant experimental feat on its own, creating this system will also help clarify a lot of thorny questions about polar vs. nematic activity, dry vs. pairwise active and substrate interactions and the role of the shape of the active agent.

This is a significant experimental advance, and based on that alone I recommend publication.

AUTHOR REPLY:

We thank the reviewer for the positive evaluation of our work.

However, I was quite disappointed with some aspects of the manuscript and I don't think it is ready for publication without more work.

First of all, the analysis of the patterns in Figures 2,3 4 and 5 is very heavily qualitative instead of quantitative.

It has been established (see e.g [22,24,25,30,40]) that the patterns found on these surfaces can be uniquely classified using their topological defects with both half and integer defects present. Here, this has only been done so (by hand) for the 'jammed' state.

AUTHOR REPLY:

We have expanded the presentation of the data to address reviewer's concerns. Yet we may want to emphasize, that the experimental limitations (due to the three dimensional geometry) in terms of contrast and bleaching make these kind of analysis extremely challenging.

Is there no way to extract a polarisation or velocity field from the data? I acknowledge that this is a very hard experiment, but it does look like the authors have 3d information at sufficient temporal resolution to try to obtain velocity information at least. In addition to

using it to look for topological defects, it could (potentially more easily) be used to find the speed of the band and vortex states, and the speed reduction in the jammed states.

AUTHOR REPLY:

We thank the reviewer for pointing out the lack of a flow/polarization field. We have now implemented an optical-flow-based analysis to extract the speed of streams and vortices more quantitatively, as shown in Figure 1.

Unfortunately, we found that the optical flow routine fails to deliver precise results in the case of the band (intensity too uniform) and of the partially/globally jammed states (where the low speed is hard to separate from fluctuations or rotations of the GUV). While it is in principle possible to extract the nematic alignment field from microscopy image, we found it to be inaccurate due to the distortions due to the spherical geometry and the available resolution, so we were forced to resort to manual tracking of defects as shown in Figure 2, which accounts nicely for the speed reduction in the jammed state. Band states instead are characterized manually and yield a speed at 40 nm/s. Additional insight about more quantitative analysis of the observed structures is given further below.

Second, I found some aspects simply unclear and it is hard to put the results here into context without more precise information:

- The mechanism of activity here is actin filaments being propelled over the inner surface of the vesicle which acts like the substrate in a motility assay. Is this correct - the manuscript is less than clear on this point? The authors need to expand on the quite cryptic 'The membrane bound [motors] non-processively propel the F-actin on the inner leaflet of the vesicle'.

AUTHOR REPLY:

We thank the reviewer for pointing out that the microscopic description of the system requires better explanations. Indeed, the mechanism is based on membrane-bound motors propelling the filaments, as introduced on fluid supported membranes recently [1]. A more accurate description of the microscopic dynamic has been added (see discussion with Reviewer 1).

Is this then really a 'dry' active interaction, i.e. non-momentum conserving, as stated in

the discussion? The inner membrane is presumably fluid, so I would expect a flow induced by the activity there as well.

AUTHOR REPLY:

By 'dry', we mean that no long-range filament–filament interaction is present, so in a sense, the system is 'dry' from the point of view of two colliding filaments. However, it is true that the fluidity of the membrane means that momentum is conserved (but adsorbed by the membrane) and that active hydrodynamic flows might be present. Given the slow dynamics of the filaments' motion, we do not expect and did not observe any indication that lipid flow could be relevant. See our comment on the possibility of motors accumulating due to diffusion given in the discussion with Reviewer 1.

We have also made the point clearer in the discussion section of the revised manuscript and together with the comment about motors' diffusion already discussed we hope this clarifies the matter.

The authors also emphasise the 'slippage of the motors on the lipid bilayer' - what is happening here?

AUTHOR REPLY:

The 'slippage of the motors' has now been clarified further to show how it leads to steric interactions between filaments. We edited the manuscript introducing a more detailed description of the microscopic behavior (see discussion with Reviewer 1) and refer to [1, 3] for further details.

- I would appreciate if the authors include more information about the experimental state: Is this system in steady-state, or does activity decrease as a function of time? There seems to be a probabilistic phase diagram (Fig 2), which would be a first for these systems. Is this truly probabilistic, or are there uncontrolled variables here, in particular activity level and vesicle radius (which is quite variable, see Fig. S3)?

AUTHOR REPLY:

We thank the reviewer for suggesting more clarity on these aspects. We have included in the revised manuscript:

”Judging from confocal movies, the system appears to be in a steady state after $\approx 15\text{--}30$ min from the beginning of the experiment and is active for at least another 30 min in the presence of an ATP-regeneration system.”

Regarding the probabilistic phase diagram, we consider Figure 2 in the manuscript a description of a general trend (from isolated structures to topology-bound ones) as the concentration is increased. Sources for variability include the experimental preparation, fluctuations in the actual concentration of actin inside a GUV with respect to the bulk one, and the size of the vesicle, which affects the effective surface concentration. For this reason the tendencies shown in Figure 2 of the original manuscript are to be considered trends rather than the points in a phase diagram.

We have now updated the Figure, as shown in Figure 3a, by adding error bars indicating the variation between different independent experiments. Additionally, we analysed the distribution of different structures as a function of the GUV’s radius (Figure 3b–e) to take this factor also into account. No clear trend is observed. In all cases, we exclude empty vesicles from the statistics.

Figure 3 has been added to the revised manuscript and Supplementary Information.

- Figure 2 and related analysis: The classification here is done purely qualitatively, and additionally ‘only vesicles showing clear patterns are considered’. As mentioned above, I am also unsure why there should be that many coexisting states at all. I don’t understand how this classification can be done unambiguously, and there are no error bars on the probability distribution plot. In particular, vortex and band states are topologically the same, and I don’t see how a clear dividing line can be drawn. Also, when does a stream become a vortex? For the (partially) jammed states, the presence of $\pm 1/2$ defect states could be a good indicator as (implicitly) done here. But it would be far better to also show that they are actually jammed, i.e. that the velocity of the material is (much) lower.

AUTHOR REPLY:

We agree with the reviewer that the classification is somewhat qualitative. While for the band and jammed states indeed we can (and do) use topological defects as an indicator, the vortex and stream phase are more arbitrary. As they do not cover the full surface of the GUV, a distinction based on the number of defects is in our opinion not feasible. For instance, double vortices have a total topological charge of $+4$ instead of displaying a total

FIG. 3. **a**, Distributions of each observed polar pattern at different encapsulated actin concentrations. Inset indicates the number n of repeated experiments and the total number N of vesicles counted at each concentration. Error bars represent the standard deviations from repeated experiments. Empty vesicles are excluded from the statistics. The graph shows a tendency towards more complex structures as the concentration increases and topological constraints become more relevant. **b–e**, The radii of vesicles at 100 nM (**b**), 150 nM (**c**), 300 nM (**d**), and 600 nM (**e**) for each observed polar pattern. The bin width is $2.5 \mu\text{m}$.

topological charge of $+2$ on the vesicle.

To clarify: at low density, we define a stream that loops on itself as a vortex whereas we define streams that hinders each other's motion as partially jammed states. The distinction between this low density states, which are all composed of streams, is rather to emphasize different aspects of the spherical geometry. Indeed, a distinction between streams (single or multiple), band (as a big stream or as a $+2$ charged-state) and jammed ($4x + 1/2$ defects) would be also acceptable, but we think it hides the different effects (confinement vs. topology) at play here.

To summarize, we have specified more in detail in the revised manuscript how the division is made:

”The fact that different polar patterns emerge with increasing c_A confirms that streams act as the building blocks for more complex structures on the closed surface. Depending on the concentration, the structures either cover only a fraction of the vesicle's surface or the

full vesicle.

Patterns that cover the full vesicle, such as bands and jammed states, must satisfy the Poincaré-Hopf theorem [11, 12] and display a total topological charge of +2. They differ in that bands display two +1 defects, and jammed states instead show multiple +1/2 defects. On the other hand, streams, vortices, and partially-jammed states are not bound by this constraint. We classify them differently only because they, while still arising from streams as fundamental blocks, display qualitatively different behaviors that have to do with confinement rather than with topology.”

In addition, the dynamics of defects in the jammed states are further discussed below.

- Figure 3 and analysis: As mentioned, a velocity field beyond the schematic picture in 3d would help a lot here. I did not find the supplementary movies associated to this mechanism particularly convincing.

AUTHOR REPLY:

We thank the reviewer for pointing out that Fig. 3 of the manuscript was not clear enough. To improve its clarity, we added a direct (manual) tracking of the stream and the +1/2 defect in the partially jammed case. Over the same observation time, the stream clearly moves much more than the jammed defect, as shown in Figure 4a&b. We also track the instantaneous speed to show this quantitatively (Figure 4c).

From this data, we can also extract the MSD (in angular coordinates) for a stream to prove that its motion is ballistic at short times (Figure 4c). In addition, we added arrows indicating the direction of motion on the equirectangular projection.

We added the following paragraph and Figure 4 to the revised manuscript:

”By the tracking of both a free moving stream and a +1/2 defect from a partially-jammed vesicle, we can visualize their trajectory and compute the instantaneous speed, as well as its mean squared displacement. This clearly shows the difference between the two configurations in terms of their motion.”

We hope this new analysis clarifies that streams move freely (Fig 3a in the manuscript), unless they end up in partially-jammed conformations where they slow down (Fig 3c in the manuscript). If the jammed configurations instead dissolves (Fig 3b in the manuscript),

FIG. 4. **a**, Trajectories of a stream moving on a vesicle. **b**, Trajectories of a $+1/2$ defect on a partially-jammed vesicle. **c**, Mean-square-displacement (MSD) of the stream in **(a)**. Dashed line indicates ballistic behavior ($\text{MSD} \propto \tau^2$). **d**, Mean-square-displacement (MSD) of the $+1/2$ defect in **(b)**. Dashed line indicates ballistic behavior ($\text{MSD} \propto \tau^2$). **e**, The instantaneous speed of stream (red) and $+1/2$ defect (blue) showing in **(a)** and **(b)**.

structures can move again.

- Figure 4 and analysis have some lovely quantitative data on the band states, and I agree with the authors that the bands share a lot of properties with their theoretical [24] and numerical [25] analogues. There seems to be a bend instability in the band in Fig. 4e - do you observe this systematically? If yes, do you have information about its wavelength?

AUTHOR REPLY:

We appreciate the reviewer's comments on our analysis. The bend instability in Fig. 4e in the manuscript is due to the equirectangular projection being performed around a north-south axis which is not perpendicular to the plane on which the band lies. In other words, if the projection is carried out along the z-axis only a ring lying on the xy-plane appears as a horizontal band in the equirectangular projection, while a tilted ring gives rise to such

skewed bands on the projection. We have re-performed the equirectangular projection along an axis for which the straightness of the band appears less ambiguous as shown in Figure 5.

FIG. 5. **a**, Equirectangular projection performed around a north-south axis which is not perpendicular to the plane on where the band lies. **b**, Equirectangular projection of the same band performed instead along a perpendicular axis. Scale bars, $10 \mu\text{m}$

Additionally, the following sentence has been added to the revised manuscript:

”Hence bands move with an approximate speed around 40 nm/s and appear to be stable, i.e., not prone to bending instabilities.”

- *Figure 5 and analysis: I appreciate the manual $+1/2$ defect tracking here, and it is certainly clear from the 5c that the dynamics is slow, with none of the oscillations seen in [22] present. Were the authors able to qualitatively establish if the defect are moving at all relative to the background? If yes, are they extensile (forward moving) or contractile (backward moving)? In simulations with (round) particles with similar mixed polar and nematic properties [30], extremely slow dynamics compared to the active motion itself was found (and it was ultimately extensile). I wonder if the present system is part of the same "activity class".*

AUTHOR REPLY:

We have included a more straight-forward tracking of defects to show that, in addition to keep their relative distance fixed, they (almost) do not move at all either, as shown in Figure 2 and explained in the discussion with Reviewer 1. Regarding the difference with the band state, we could estimate a speed of 40 nm/s for the band by manual observation, which

clearly is higher than the jammed state. We have stated this explicitly (see above).

- [1] Sciortino, A. & Bausch, A. R. Pattern formation and polarity sorting of driven actin filaments on lipid membranes. *Proc Natl Acad Sci U S A* **118** (2021).
- [2] Grossmann, R., Aranson, I. S. & Peruani, F. A particle-field approach bridges phase separation and collective motion in active matter. *Nat Commun* **11**, 5365 (2020).
- [3] Grover, R. *et al.* Transport efficiency of membrane-anchored kinesin-1 motors depends on motor density and diffusivity. *Proc Natl Acad Sci U S A* **113**, E7185–E7193 (2016).
- [4] Weitz, S., Deutsch, A. & Peruani, F. Self-propelled rods exhibit a phase-separated state characterized by the presence of active stresses and the ejection of polar clusters. *Phys. Rev. E* **92**, 012322 (2015).
- [5] Huber, L., Suzuki, R., Kruger, T., Frey, E. & Bausch, A. R. Emergence of coexisting ordered states in active matter systems. *Science* **361**, 255–258 (2018).
- [6] Memarian, F. L. *et al.* Active nematic order and dynamic lane formation of microtubules driven by membrane-bound diffusing motors. *Proc Natl Acad Sci U S A* **118** (2021).
- [7] Sknepnek, R. & Henkes, S. Active swarms on a sphere. *Phys Rev E* **91**, 022306 (2015).
- [8] Shankar, S., Bowick, M. J. & Marchetti, M. C. Topological sound and flocking on curved surfaces. *Phys. Rev. X* **7**, 031039 (2017).
- [9] Brandstätter, T. *et al.* Curvature induces active velocity waves in rotating multicellular spheroids. (2021). arXiv:2110.14614.
- [10] Henkes, S., Marchetti, M. C. & Sknepnek, R. Dynamical patterns in nematic active matter on a sphere. *Phys Rev E* **97**, 042605 (2018).
- [11] Kamien, R. D. The geometry of soft materials: a primer. *Rev. Mod. Phys.* **74**, 953–971 (2002). URL <https://link.aps.org/doi/10.1103/RevModPhys.74.953>.
- [12] Frankel, T. *The Geometry of Physics: An Introduction* (Cambridge University Press, 2011), 3 edn.

REVIEWERS' COMMENTS

Reviewer #1 (Remarks to the Author):

I would like to thank the authors for their reply and the implemented changes in the manuscript. I recommend the publication of the manuscript in its current form.

Reviewer #2 (Remarks to the Author):

The authors have answered all comments and questions I had.

Reviewer #3 (Remarks to the Author):

I would like to thank the authors for the extra work they have put into the manuscript. I would also like to apologise for the tone of my first review, unfortunately I have been under a lot of stress lately.

The new optical flow and manual measurements of the flow velocity really do help the manuscript. It is now clear to the reader that the jammed states are moving very slowly if at all, and that the partially jammed states are distinct. I do appreciate that these are very hard experiments. Supplementary figures 5&6 are very nice, and would equally have been at home in the manuscript itself.

Equally, I appreciate the 'stochastic phase diagram' situation better. As shown in the new supplementary figure 4, actin concentration is the main variable determining the pattern, while radius has a sub-dominant influence. I still wonder if the phase diagram truly is stochastic, but that's a question for follow-up experimental or numerical work.

Finally, thank you for including more detail about the active motion mechanism of the filaments. I do realise that it's been published before, but it does help make the manuscript more accessible to the non hyper-specialised reader.

This is ready to be published as far as I am concerned. My only request is that the authors strongly consider promoting some of the SI figures to the main text (to my taste, not all the experimental images and schematic drawings there are strictly necessary). In particular, figures S4c-d, S5a-b, and S6a-c really do add to the story.

We thank the reviewers for the careful reading of our manuscript and for the insightful comments, which we address in detail below. We report the original comments in italics and add our response in normal font following the phrase "AUTHOR REPLY".

Reviewer: 1

Comments:

I would like to thank the authors for their reply and the implemented changes in the manuscript. I recommend the publication of the manuscript in its current form.

AUTHOR REPLY:

We thank the Reviewer for the positive evaluation of our work and the recommendation for publication.

Reviewer: 2

Comments:

The authors have answered all comments and questions I had.

AUTHOR REPLY:

We thank the Reviewer for the positive evaluation of our work and the recommendation for publication.

Reviewer: 3

Comments:

I would like to thank the authors for the extra work they have put into the manuscript. I would also like to apologise for the tone of my first review, unfortunately I have been under a lot of stress lately.

The new optical flow and manual measurements of the flow velocity really do help the

manuscript. It is now clear to the reader that the jammed states are moving very slowly if at all, and that the partially jammed states are distinct. I do appreciate that these are very hard experiments. Supplementary figures 5&6 are very nice, and would equally have been at home in the manuscript itself.

Equally, I appreciate the 'stochastic phase diagram' situation better. As shown in the new supplementary figure 4, actin concentration is the main variable determining the pattern, while radius has a sub-dominant influence. I still wonder if the phase diagram truly is stochastic, but that's a question for follow-up experimental or numerical work.

Finally, thank you for including more detail about the active motion mechanism of the filaments. I do realise that it's been published before, but it does help make the manuscript more accessible to the non hyper-specialised reader.

This is ready to be published as far as I am concerned. My only request is that the authors strongly consider promoting some of the SI figures to the main text (to my taste, not all the experimental images and schematic drawings there are strictly necessary). In particular, figures S4c-d, S5a-b, and S6a-c really do add to the story.

AUTHOR REPLY:

We thank the Reviewer for the positive evaluation of our work and the recommendation for publication. We have included Supplementary Figure 5&6 now in the manuscript to enhance the presentation of our work.